# Omega-3 Fatty Acids, Cognition, and Brain Volume in Older Adults

**DOI:** 10.3390/brainsci13091278

**Published:** 2023-09-02

**Authors:** Spencer Loong, Samuel Barnes, Nicole M. Gatto, Shilpy Chowdhury, Grace J. Lee

**Affiliations:** 1Department of Psychology, School of Behavioral Health, Loma Linda University, Loma Linda, CA 92350, USA; sloong@students.llu.edu; 2Department of Radiology, School of Medicine, Loma Linda University, Loma Linda, CA 92350, USA; sabarnes@llu.edu (S.B.);; 3School of Public Health, Loma Linda University, Loma Linda, CA 92350, USA; ngatto@llu.edu

**Keywords:** aging, diet, omega-3 fatty acids, brain volume, cognition, memory

## Abstract

The elderly population is growing at increased rates and is expected to double in size by 2050 in the United States and worldwide. The consumption of healthy foods and enriched diets have been associated with improved cognition and brain health. The key nutrients common to many healthy foods and diets are the omega-3 polyunsaturated fatty acids (omega-3 FAs), such as eicosapentaenoic acid (EPA) and docosahexaenoic acid (DHA). We explored whether omega-3 FA levels are associated with brain volume and cognition. Forty healthy, cognitively normal, Seventh-day Adventist older adults (mean age 76.3 years at MRI scan, 22 females) completed neurocognitive testing, a blood draw, and structural neuroimaging from 2016 to 2018. EPA and an overall omega-3 index were associated with individual measures of delayed recall (RAVLT-DR) and processing speed (Stroop Color) as well as entorhinal cortex thickness. EPA, DHA, and the omega-3 index were significantly correlated with the total white matter volume. The entorhinal cortex, frontal pole, and total white matter were associated with higher scores on delayed memory recall. This exploratory study found that among healthy, cognitively older adults, increased levels of omega-3 FAs are associated with better memory, processing speed, and structural brain measures.

## 1. Introduction

The aging population is expected to double from 703 million individuals 65 years or older to 1.5 billion globally by 2050 [1]. The symptoms associated with aging include decreased physical abilities, a loss of muscle mass and strength, changes in brain structure and function, and decreased cognitive abilities [2,3,4,5]. The processing speed and efficiency, encoding of new memories, working memory, and reasoning skills have been observed to decline starting in early adulthood, while crystallized knowledge, such as vocabulary or general information, appears to remain relatively stable until the 60s [6,7]. Many factors influence the trajectory of one’s physical and cognitive abilities, including greater physical activity, more social contacts, the absence of depression and cognitive impairment, and nutrition [8]. 

Some of the physiological benefits of a healthy diet have been well documented and include a decreased risk of cardiovascular disease, coronary heart disease, stroke, diabetes, dyslipidemias, hypertension, and obesity [9,10,11]. In recent years, the benefits of a healthy diet on brain function have also been explored. The Mediterranean diet (MeDi), the Dietary Approaches to Stop Hypertension (DASH) diet, and the Mediterranean-DASH Intervention for Neurodegenerative Delay (MIND) diet have all received considerable research and have been found to have positive effects on both brain function and structure [12,13,14,15,16,17,18]. 

Omega-3 fatty acids (FAs) are common elements in the MeDi, DASH, and MIND diets. Foods that contain high amounts of omega-3 FAs include vegetable oils, flaxseeds, walnuts, vegetables, and fatty fish [19,20]. The accumulating evidence from many studies indicates that increased consumption of omega-3 FAs may serve as a protective factor against the negative consequences of aging [21,22,23,24,25,26]. Specifically, omega-3 FAs have been associated with positive effects on both cognition [27,28,29,30] and various brain measures, including hippocampal, gray matter, and total brain volumes and white matter microstructural integrity [18,31,32]. Moreover, a diet rich in omega-3 FAs was found to be correlated with increased synaptic density [21] and decreased neuronal loss [22]. 

Further studies have found an inverse relationship between fish intake and cognitive impairment [33]. Working memory and selective attention improvements were noted after five weeks [34] and executive functions after six months of fish oil supplementation [18]. A lower omega-3 index (DHA + EPA) was associated with lower scores for visual memory, executive function, and abstract thinking [35]. Nevertheless, other studies have not found associations between omega-3 FA supplementation and cognition [36,37], and some randomized controlled trials have not documented a significant change in global cognitive function or language, revealing only small improvements in executive function, memory, and visuospatial skills [38,39,40]. These discrepant findings highlight the importance of further exploring the cognitive and neurophysiological effects of omega-3 FAs. 

The current study examines the associations between blood serum levels of omega-3 FAs, cognitive function, and brain volume to explore the relationship between omega-3 FAs and brain health among a sample of older participants of the Adventist Health Study-2 (AHS-2). Members of the Seventh-day Adventist (SDA) church typically lead a relatively healthy and active lifestyle and adhere to healthy eating patterns (e.g., follow a vegetarian or vegan diet) and abstain from smoking and drinking alcohol, which have been associated with brain atrophy and cognitive deficiencies [41,42,43]. 

## 2. Materials and Methods

### 2.1. Study Population

The Adventist Health Study-2 (AHS-2), a longstanding prospective cohort study of over 96,000 Seventh-day Adventists in the United States and Canada was established in 2002 to explore the associations between diet, lifestyle, and health outcomes [44,45]. In 2016, 2685 members of the cohort were identified who were 60 years or older, community-dwelling, and living within 75 miles of Loma Linda University (LLU). From 2016 to 2018, 916 (34.1%) were randomly identified, among which 199 (21.7%) were reached by telephone and invited to participate in the AHS-2 Cognitive and Neuroimaging (AHS-2CAN) substudy. Of those, 168 agreed to participate and were screened for eligibility. The exclusion criteria included not being able to understand or speak English proficiently or having any acute medical conditions that could negatively impact cognitive function. One hundred and thirty-two participants who met the inclusion criteria were enrolled in the AHS2-CAN substudy to investigate the effect of dietary patterns on cognitive aging, and they completed baseline study procedures, including cognitive and physical assessments. 

Approximately 1 year after enrolling in the AHS2-CAN, the participants were invited to return for a follow-up cognitive assessment as well as a brain MRI scan. Additional exclusions were made for conditions that would be contraindicated for a brain MRI with contrast: pacemakers or other implanted devices; a history of kidney disease or diabetes; and claustrophobia [46]. Forty-five AHS2-CAN participants (34.1%) completed the follow-up assessment, of whom 40 completed the MRI scan. Five participants were unable to complete the MRI for the following reasons: one had metal in his/her body, one was unable to lie flat for the purpose of MRI acquisition, and three declined due to symptoms of claustrophobia or concerns related to the contrast agent. 

All the study procedures were approved by the LLU Institutional Review Board, and written informed consent was obtained from all the participants. 

### 2.2. Neuropsychological Tests

The participants’ performance on tests of memory, processing speed, attention and working memory, and executive function were assessed at the time of the MRI and blood collection. The participants completed the Rey Auditory Verbal Learning Test (RAVLT), a test of verbal learning and memory [47]; the RAVLT Immediate Recall score (RAVLT-IR, total words recalled across the first 5 learning trials) and Delayed Recall score (RAVLT-DR, total words recalled after a 30 min delay) were examined. In the Stroop Test [48], the Stroop Word and Stroop Color measure processing speed, as reflected by the total number of words read and colors named, respectively, within each 45 s trial; the Stroop Color/Word Inhibition measures executive function (i.e., selective attention and response inhibition) as reflected by the number of mismatched color word/ink combinations correctly completed in 45 s. The Digit Span subtest of the Wechsler Adult Intelligence Scale—4th Edition (WAIS-IV) [49] was included as a measure of attention and working memory; the total raw score (sum of number sequences correctly repeated in forward, backward, and sequencing order) was examined. Cognitive testing was conducted by trained study personnel. 

### 2.3. Omega-3 Fatty Acids

A drop of blood was collected on filter paper that was pre-treated with an antioxidant cocktail (Fatty Acid Preservative Solution, FAPS™) and allowed to dry at room temperature for 15 min. The dried blood spots (DBS) were immediately stored in a −80 °C freezer until they were shipped to OmegaQuant for FA analysis. Eicosapentaenoic acid (EPA) and docosahexaenoic acid (DHA) were measured, and an omega-3 index was calculated as the sum of the EPA and DHA, adjusted by a regression equation (r = 0.97) to predict the omega-3 index in red blood cells (RBC) (see Appendix A). The FA composition was expressed as a percent of the total identified fatty acids. The RBC FA composition has been shown to reflect FA intake up to 120 days prior to measurement compared to plasma concentrations, which reflect the intake over the last few days [35,46,50]. 

### 2.4. MRI Acquisition and Image Analysis

MR imaging was performed at Loma Linda University Medical Center on an existing 3T Siemens Skyra (Siemens Medical Systems, Erlangen, Germany) using a 32-channel array head coil to provide greater signal-to-noise and parallel imaging capability to facilitate faster imaging. Multiple sequences were acquired in the 45 min of scan time: 3D T1-weighted MPRAGE, Dynamic Contrast Enhanced (DCE), Diffusion Tensor Imaging (DTI), Susceptibility Weighted Imaging (SWI), and FLAIR. 3D T1-weighted MPRAGE images with the sequencing parameters TR/TE = 1950/2.3 ms, TI = 900 ms, FA = 8°, FOV = 240 × 240, matrix 256 × 256, GRAPPA = 2, acquisition time = 4:30, and slice thickness = 0.9 mm were used for the analyses.

The image analysis consisted of cortical/subcortical volumetric segmentation, cortical surface reconstruction, and parcellation of the T1-weighted scans using the FreeSurfer software package version 7.2.0 [51,52,53,54,55,56]. FreeSurfer is an open-source software package for the automatic processing and analysis of brain MRI images, and it has been found to produce comparable results [57,58], or in some cases more accurate results [59], than manual tracing. To generate the PFC thickness (frontal pole) and the volumes for the hippocampus, entorhinal cortex, and total white matter, the FreeSurfer command recon-all was used. 

### 2.5. Statistical Analysis

The descriptive statistics of the participants were summarized. Two-tailed, partial correlations were used to calculate *p*-values. A statistically significant correlation was considered at *p* < 0.05. Partial correlations, adjusted for age, sex, and education as continuous variables between the omega-3 FAs (EPA, DHA, omega-3 index), brain volume and thickness (hippocampus, entorhinal cortex, and frontal pole), and cognitive scores were calculated. SPSS v.28 [60] was used for all the statistical analyses.

## 3. Results

### 3.1. Demographic and Clinical Characteristics of Study Participants

The mean age of the participants (*n* = 40) was 76.3 years (SD = 8.3, range = 63–90 years), with an average of 16.8 years of education (SD = 2.5) (Table 1). Thirty-four participants were non-Hispanic White (85.0%), one was Black or African American (2.5%), two were Asian (5.0%), two were Hispanic (5.0%), and one was Native Hawaiian or Pacific Islander (2.5%). 

### 3.2. Correlations between Omega-3 Fatty Acids and Cognitive Scores

After controlling for age, sex, and education, multiple statistically significant correlations were observed between the cognitive scores and omega-3 FAs (Table 2). EPA was significantly positively correlated with RAVLT-DR (r = 0.40, *p* = 0.02) and Stroop Color (r = 0.50, *p* = 0.003). Statistically significant positive correlations were observed between the omega-3 index and RAVLT-DR (r = 0.38, *p* = 0.03) and Stroop Color (r = 0.39, *p* = 0.02). No statistically significant correlations were observed between the omega-3 variables and RAVLT-IR, Stroop Word, or Digit Span.

### 3.3. Correlations between Omega-3 Fatty Acids and Brain Region Volumes and Thickness

After controlling for age, sex, and education, statistically significant positive correlations were found between EPA and EC (r = 0.41, *p* < 0.05) and the omega-3 index and EC volumes (r = 0.34, *p* < 0.05) (Table 3). EPA, DHA, and the omega-3 index were all significantly positively correlated with white matter volume (all *p* < 0.05).

### 3.4. Correlations between Brain Volume and Thickness and Cognitive Scores

The entorhinal cortex volume was statistically significantly correlated with RAVLT-DR (r = 0.36, *p* < 0.04) (Table 4). The frontal pole thickness was significantly positively correlated with RAVLT-IR (r = 0.45, *p* = 0.01) and RAVLT-DR (r = 0.36, *p* = 0.04). The white matter volume was significantly positively correlated with RAVLT-DR (r = 0.36, *p* = 0.04). No statistically significant correlations were observed with Stroop or Digit Span.

## 4. Discussion

The results of this exploratory cross-sectional study demonstrate that among a healthy, cognitively normal, aging population, omega-3 FA levels had variable associations with cognition and the brain region volume and thickness as assessed by MRI. Significant correlations were observed between both EPA and the omega-3 index and measures of delayed memory and processing speed but not with measures of working memory and executive function. EPA and the omega-3 index were also correlated with the entorhinal cortical volume, and all three omega-3 FA variables were correlated with the total white matter volume. None of the omega-3 FA variables were correlated with the hippocampal volume or the frontal pole cortical thickness.

These findings are generally consistent with previous studies showing associations between levels of omega-3 FAs, cognitive functioning, and brain regions of interest [18,21,35,61]. Other studies [36,37] did not find associations between omega-3 FAs and cognition. Two features make our study unique and may offer some explanation for how our findings compare to those of other studies. First, methodological differences in omega-3 FA measurement (e.g., blood, plasma, or serum) may at least partially explain discrepant findings and make study comparisons more difficult. For the current study, we utilized a serum measurement of omega-3 FAs. This method estimates FA intake up to 120 days and reflects longer-term FA levels compared to other methods that measure acute levels. Gatto et al. [45] also found that this study cohort exhibited a stable and consistent dietary pattern over their lifetime. Therefore, not only did our FA measurement reflect intake over a longer period but the study participants were also more likely to have consistently adhered to the same diet over time. Other methodological differences that have been noted in the literature include subgroup effects (e.g., genetic or risk factor burden), short treatment periods, or dosing and tracking [62].

Due to the documented effects of omega-3 FAs on synaptic membranes, preventing neuronal loss and atrophy, and other brain health and functioning [21,63,64], we expected the results to indicate stronger associations between omega-3 FA levels and the regions of interest, particularly the subcortical regions, namely the hippocampus. Although the results did not indicate statistically significant associations between the omega-3 FA levels and the hippocampus, statistically significant associations between EPA and the omega-3 index and the entorhinal cortical volume were observed. The entorhinal cortex is largely known as the major input and output structure of the hippocampus and is involved with learning and memory [65,66,67]. Due to the physiological makeup of the entorhinal cortex, namely the white matter tracts connecting to the hippocampal formation and other cortical projections, it may be possible that the entorhinal cortex is more sensitive to the effects of omega-3 FAs on white matter and other neuronal properties.

Additionally, we observed statistically significant associations between all the omega-3 FA variables and the white matter volume. The white matter microstructure has been found to mediate the relationship between a dietary pattern high in omega-3 FAs and cognitive scores [68]. The effect of omega-3 FAs on the white matter microstructure may be due to decreased inflammation or oxidative stress or their role in pathways directly involving axonal loss or demyelination. Another possible mechanism underlying the relationship between omega-3 FAs and white matter integrity in older adults, in particular, is that omega-3 FAs may help preserve the integrity of fibers that are particularly vulnerable to aging and, subsequently, slow white matter degeneration [69]. Although there are some contradictory findings [70,71], the association between increased omega-3 FA intake and white matter volume appears consistent with other studies that revealed that omega-3 FAs increase the white matter volume or microstructural integrity [35,72,73,74,75].

Omega-3 FAs are critical components of brain structure and functioning, with the effects reported to include increased gray matter volume and white matter microstructural integrity [18,31] and improved cognitive function [27,28,29,30]. Omega-3 FAs exert their influence on cellular components (e.g., proteins, receptors, ion channels, and enzymes), which can lead to decreased inflammation, increased synaptic plasticity and dendritic spines, and improved neurotransmission and signaling [21,22,23,73]. The additional effects of omega-3 FAs may reduce the production of amyloid-β, implicated in Alzheimer’s disease, and increase brain-derived neurotrophic factor (BDNF) and synaptic protection [35,76,77].

These results should be interpreted while considering the following limitations. First, the small sample size limited our statistical power. However, the effect sizes observed in this study ranged from small to medium and, in general, the correlations were trending in expected directions. The authors cannot determine causality and are unable to assess atrophy rates in the brain. The entorhinal cortex, or rather atrophy of the entorhinal cortex, appears to be particularly sensitive to poorer memory performance and pathological processes, including the conversion from mild cognitive impairment to Alzheimer’s disease [78,79,80,81]. Finally, given our small sample size, which limited the complexity of our statistical analysis, we could not adjust the correlations for several factors that have been shown to affect the metabolism and absorption of omega-3 FAs [82,83,84,85,86,87] (e.g., genetic factors, physical activity and health status, smoking, alcohol consumption, metabolic syndromes, and geography). Nevertheless, the health-conscious lifestyle practices of our study population, including abstaining from smoking and alcohol and remaining physically active, reduce the likelihood that these factors could confound the associations we observed.

This study has many strengths. As noted, the unique sample of SDA participants is advantageous in that it provides a fairly homogenous group with fewer confounding variables (e.g., diverse lifestyles and geographic location). However, the omega-3 FA levels measured in our sample were slightly lower than other studies that used similar measurement techniques [18,35,88], which may be due to the unique characteristics of the participants. SDA church members are less likely to consume oily fish, which is one of the more highly concentrated sources of long-chain omega-3 FAs, especially EPA, thereby likely lowering their overall percentage of EPA levels and the omega-3 index. Gu et al. [13] noted that the association they found between adherence to a Mediterranean-type diet and larger grey and white matter volumes was likely driven by high fish and low meat consumption. Furthermore, the study sample being drawn from an SDA population may decrease the generalizability of our findings to the general population. Approximately two-thirds of our sample participants reported following a vegetarian or vegan type of diet, which is much higher than the approximately 6% in the US population in 2022 [89]. However, even with these decreased omega-3 FA levels in our sample, we still observed statistically significant associations between the variables.

## 5. Conclusions

Lifestyle factors are of interest as a means of preventing or delaying cognitive decline. Diet and nutrition, in particular, have garnered attention due to their low cost to the general population and reported neuroprotective effects. In this study, we investigated the relationships between the levels of red blood cell omega-3 FAs, brain volume, and cognition. We found that higher levels of omega-3 FAs were associated with better performance on tests of memory and processing speed and greater brain volume in the entorhinal cortex and total white matter.

More research is required to determine the extent to which omega-3 FA consumption affects cognitive ability and/or protects against or delays cognitive decline. Recommendations for future studies investigating the effects of fatty acids on brain function include utilizing additional forms of neuroimaging (e.g., fMRI, DTI), larger sample sizes, and longitudinal studies. The results of this exploratory cross-sectional study provide support for the associations between omega-3 FAs, regions of interest, and cognitive abilities in a healthy, cognitively normal, older adult population.

## Figures and Tables

**Table 1 brainsci-13-01278-t001:** Descriptive information of study sample (*n* = 40).

Variable	*n*	%
Sex		
Female	22	55.0
Male	18	45.0
Age (Mean ± SD (range))	76.3 ± 8.3 (63–90)	
60–69	10	25.0
70–79	15	37.5
80–89	13	32.5
90–96	2	5.0
Years of Education (Mean ± SD (range))	16.8 ± 2.5 (10–20)	
≤12	1	2.5
13–16	19	47.5
17–18	10	25.0
≥19	10	25.0
Race		
Caucasian	34	85.0
Black or African American	1	2.5
Asian	2	5.0
Hispanic	2	5.0
Native Hawaiian or Pacific Islander	1	2.5
Marital Status		
Married	29	72.5
Widowed	5	12.5
Single/Never Married	3	7.5
Divorced	3	7.5
Family History of Neurological Illness		
None	25	62.5
Stroke	4	10.0
Dementia	2	5.0
Alzheimer’s Disease	4	10.0
Parkinson’s Disease	1	2.5
Other	4	10.0
Omega-3 Fatty Acids	Mean ± SD (range)	
EPA	0.9% ± 0.7% (0.15–3.45)	
DHA	2.6% ± 1.1% (1.14–5.74)	
Omega-3 Index	5.4% ± 1.9% (2.88–11.79)	

**Table 2 brainsci-13-01278-t002:** Correlations between omega-3 fatty acids and cognitive scores.

	Immediate Memory	Delayed Memory	Processing Speed		Executive Functions	
Fatty Acid	RVLT-IR	RAVLT-DR	Stroop Word	Stroop Color	Stroop C/W	Digit Span
EPA	−0.07	0.40 *	0.23	0.50 **	0.32 ^†^	0.18
DHA	0.03	0.33 ^†^	0.22	0.29	0.19	0.06
Omega-3 Index	−0.01	0.38 *	0.24	0.39 *	0.26	0.11

Note: All values are Pearson *r* coefficients, controlled for age (years; continuous), sex (male, female), and education (years; continuous). RAVLT = Rey Auditory Verbal Learning Test; IR = Immediate Recall; DR = Delayed Recall; Stroop C/W = Stroop Color/Word Inhibition ** p* < 0.05, ** *p* < 0.01, ^†^ 0.05 < *p <* 0.10.

**Table 3 brainsci-13-01278-t003:** Correlations between omega-3 fatty acids and brain region volumes and thickness.

	Fatty Acid		
Brain Region	EPA	DHA	Omega-3 Index
Hippocampus	0.16	0.04	0.09
Entorhinal Cortex	0.41 *	0.26	0.34 *
Frontal Pole	−0.22	−0.12	−0.17
White Matter	0.34 *	0.33 *	0.36 *

Note: All values are Pearson *r* coefficients, controlled for age, sex, and education. * *p* < 0.05.

**Table 4 brainsci-13-01278-t004:** Correlations between brain region volumes and thickness and cognitive scores.

	Immediate Memory	Delayed Memory	Processing Speed		Executive Functions	
Brain Region	RAVLT-IR	RAVLT-DR	Stroop Word	Stroop Color	Stroop C/W	Digit Span
Hippocampus	0.22	0.33 ^†^	−0.25	−0.13	−0.22	−0.07
Entorhinal Cortex	0.31 ^†^	0.36 *	−0.10	0.13	0.18	−0.10
Frontal Pole	0.45 *	0.36 *	−0.20	−0.06	−0.14	−0.08
White Matter	0.24	0.36 *	0.01	0.02	0.01	0.06

Note: All values are Pearson *r* coefficients, controlled for age, sex, and education. RAVLT = Rey Auditory Verbal Learning Test; IR = Immediate Recall; DR = Delayed Recall; Stroop C/W = Stroop Color/Word Inhibition. ** p* < 0.05, ^†^ 0.05 < *p* < 0.10.

## Data Availability

The data presented in this study are available upon request from the corresponding author.

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
