# Peer review of "Omega-3 Fatty Acids, Cognition, and Brain Volume in Older Adults"

_brainsci, 2023, doi:10.3390/brainsci13091278_

Round 1

Reviewer 1 Report

The manuscript investigates the association between omega-3 FA levels, neurocognitive performance, and structural brain measures. The topic is highly relevant given the growing elderly population and the increasing interest in the potential benefits of healthy diets for brain health. Overall, the manuscript presents promising findings and contributes valuable insights into the association between omega-3 FA levels and brain health in older adults. However, several areas require clarification and additional analysis to strengthen the manuscript.

(1)The author's comprehensive literature review has brought to light discrepant findings in previous research. The intriguing question arises as to whether the current study's results can offer insights that may explain the reasons behind the discrepancies observed in earlier studies. If the present study fails to provide a definitive explanation for the discrepancies in previous research, it may be considered merely one of the discrepant studies, with limited contribution in terms of resolving the inconsistency in the literature.

(2)Selecting 135 individuals from a pool of 96,000 subjects solely based on the criteria of English proficiency and good physical health does  raise questions about the rationale and practical feasibility of such a selection process.

(3) The concentration of Omega-3 fatty acids in blood can be influenced by several factors. Diet plays a crucial role, as the intake of Omega-3 fatty acids through foods like fish, nuts directly affects blood levels. Lifestyle factors, such as physical activity, weight control, smoking, and alcohol consumption, may also impact Omega-3 levels. Genetic factors can affect an individual's ability to synthesize and metabolize Omega-3 fatty acids. Additionally, age and gender may play a role. An individual's health status, as certain conditions like cardiovascular disease, inflammatory disorders, and metabolic syndrome can influence Omega-3 metabolism and absorption. As the author's study is based on blood Omega-3 fatty acid levels, it is essential to emphasize the limitations of the research findings. 

Reviewer 2 Report

The manuscript by Loong et al. describes the exploratory cross-sectional research on the correlations between levels of omega-3 polyunsaturated fatty acids, particularly EPA and DHA, with cognition and specified brain parts volume. The Authors found some positive correlations, supporting growing knowledge about the beneficial actions of the studied molecules. The study is well-planned and described and will probably be interesting for readers of Brain Sciences. The reviewer has only two minor suggestions:

1. The Appendix mentioned in line 119 was unavailable for review. Please ensure it is included in the final version of the article.

2. The reviewer suggests adding (most probably as supplementary material) plots with data used for the calculation of all shown correlations.

Round 2

Reviewer 1 Report

The author has addressed the raised concerns adequately.